

# Evaluation of knowledge, attitudes, and clinical education of dental students about COVID-19 pandemic

Osman Ataş[1] and  Tuba Talo Yildirim[2]

[1] Department of Pediatric Dentistry, Faculty of Dentistry, Firat University, Elazig, Turkey
[2] Department of Periodontology, Faculty of Dentistry, Firat University, Elazığ, Turkey

Corresponding author
Osman Ataş,
osman_atas88@hotmail.com,
o.atas@firat.edu.tr

## ABSTRACT

**Background**. The novel coronavirus disease (COVID-19) is a new viral respiratory illness, first identified in Wuhan province, China. Dental professionals and dental students are at an increased risk for these viruses from dental patients, as dental practice involves face-to-face communication with the patients and frequent exposure to saliva, blood, and other body fluids. Dental education can play an important role in the training of dental students, adequate knowledge and adopting attitudes regarding infection control measures. The aim of this study was to evaluate knowledge, attitudes, and clinical education of dental students about COVID-19 pandemic.

**Methods**. A total of 355 pre-clinical and clinical dental students (242 and 113, respectively, comprising 190 females and 165 males) at Fırat University Dentistry Faculty, in Elazığ, Turkey answered an online questionnaire about the biosafety procedures for and their attitudes to and knowledge of COVID-19. The study was conducted in March 2020, Turkey. The data gained were analyzed using descriptive statistical methods and chi-square test.

**Results**. Both the clinical and preclinical students were found to be afraid of infecting themselves and their environment with COVID-19, and the difference between them was statistically significant. Three quarters (74.9%) of the participants responded yes to the question of whether they thought that experiences related to COVID-19 affected them psychologically, with the differences between gender and clinical status were statistically significant. Responses to the question of which clinical rotation worried them more were 29.9% endodontics, 25.1% oral and maxillofacial surgery, 16.3% prosthesis, 15.2% periodontology, 6.8% restorative dentistry, 3.9% oral diagnosis and radiology, 1.7% pedodontics, and 1.1% orthodontics, with a significant difference between the preclinical and clinical students. Regarding the measures applied by the clinical students in their clinical rotation, the responses were 100% gloves and 100% mask (with 11.5% FFP3/N95 mask), 73.6% face protective shield and 37.1% safety glasses, and 49% bonnet and 16.8% disposable box, with 90.2% frequent hand washing, and 86.7% frequent hand antiseptic usage.

**Conclusions**. While students gave good responses regarding the standard measures they take to protect against transmission of COVID-19, their knowledge and attitudes about the extra measures they can take should be improved. For students to be least affected by fears associated with the disease, dental faculties should be ready to provide psychological services to those in need.

## INTRODUCTION

An outbreak of pneumonia with an unknown etiology occurred in December 2019 in Wuhan, China (*Ge et al., 2020*). A month later, scientists isolated a new coronavirus (SARS-CoV-2), which was found to cause severe acute respiratory syndrome (*Meng, Hua & Bian, 2020*). The pathogen was identified as the seventh member of the coronavirus family to have infected humans, and the disease it caused became known as the 2019 corona virus disease, or COVID-19 (*Ge et al., 2020*; *Peng et al., 2020a*). COVID-19 created a public health problem affecting not only China but the whole world. On January 31, 2020 the World Health Organization (WHO) declared COVID-19 an international emergency that threatened public health. Later, the infection became much more widespread, and on March 12, 2020 the WHO updated the situation, declaring COVID-19 a pandemic.

A coronavirus is a type of virus that can be transmitted from animals to humans; in such cases, this virus mutates when it passes to humans, further leading to human-to-human spread (*Chan et al., 2020b*). A coronavirus can progress in different stages, such as mild, moderate, and severe, and such viruses are in the same group as Severe Acute Respiratory Syndrome (SARS) of 2002 and Middle East Respiratory Syndrome (MERS) of 2012 (*Huang et al., 2020*). The new coronavirus is generally a disease that manifests in symptoms of high fever and cough, and in advanced cases, patients may endure respiratory distress. In addition, it has been shown that different symptoms such as nausea, vomiting, diarrhea, muscle-joint pain, and loss of appetite may occur. In severe cases, pneumonia, severe respiratory failure, kidney failure, and death may occur (*Sabino-Silva, Jardim & Siqueira, 2020*). A coronavirus is a type of virus that can be transmitted from animals to humans; in such cases, this virus mutates when it passes to humans, further leading to human-to-human spread (*Phan et al., 2020*).

COVID-19 is detected in the saliva of infected patients, so dental/oral and other healthcare professionals in particular should be very careful in protecting against the spread of the disease (*Sabino-Silva, Jardim & Siqueira, 2020*; *To et al., 2020*). Transmission is similar to other respiratory diseases; it can occur with droplets ejected during speaking, coughing, or sneezing (activities of the respiratory system) and also through aerosols employed during clinical procedures (*Sabino-Silva, Jardim & Siqueira, 2020*).

In this process, dentists may provide routes for virus transmission from unrecognized COVID-19-infected patients and patients under surveillance. It appears possible to have asymptomatic infections. Thus, the contamination may occur before symptoms of the disease appear. Relatedly, a recent clinical study showed that 29% of 138 COVID-19 patients hospitalized in Wuhan, China were healthcare professionals (*Chan et al., 2020a*).

Routine dental practices that emit aerosols pose a risk to patients, dentists, and auxillary staff (*Sabino-Silva, Jardim & Siqueira, 2020*). As with bronchoscopy, using aerosols during dental treatments may constitute a high-risk procedure for these people in respect of the

inhalation of airborne particles, ocular transmission, causing them to be directly exposed to the virus (*Leonard et al., 2020*).

Therefore, dentists and dental students need to be very careful and showed develop preventive strategies to avoid COVID-19 involving, for example, hand hygiene, personal protective equipment (PPE) and cross contamination prevention methody for all staff when performing aerosol-emitting procedures. It is inevitable that dentistry faculty students with insufficient clinical experience will be more exposed to infectious diseases (*Stewardson et al., 2002*). In previous studies, occupational exposure to infective diseases in dental faculties has been reported as 66–80% (*Kennedy & Hasler, 1999*; *Stewardson et al., 2002*).

In order to increase the compliance of dental students with universal precautions and to eliminate their deficiencies, students understanding and behavior should be determined. In this study, students in dental school setting were questioned in order to evaluate their general knowledge levels, attitudes, and practices in regard to COVID-19.

## MATERIAL AND METHODS

The cross-sectional study was conducted at the Faculty of Dentistry at Firat University in March 2020, during the week after the first reported COVID-19 cases in Turkey. The participants were pre-doctoral dental students performing their preclinical education (first, second, and third classes) and clinical rotations (fourth and fifth classes). An online questionnaire was developed in Google Forms (Elazig, Turkey) containing 17 questions about the students' knowledge, attitudes, and practices of pre-doctoral students in respect of this new disease COVID-19. The study was approved by Institu Review Board of the Firat University Research Ethics Committee (2020/30-06). All participants voluntarily participated in this study. Participants were informed about the nature of the study. The IRB did not request written informed consent form due to being a cross-sectional study where no personal identifiers were used. Prepared e-survey forms were sent to students via a link created for the purpose. It was explained at the beginning of the questionnaire that the purpose of the data collection was for scientific research. From the total number of 363 students in the dental school, 355 completed the whole questionnaire (response rate 97.7%). When we conducted this questionnaire (the first week of the pandemic in Turkey), the students continued their education. With increased COVID-19 outbreak in Turkey, education was interrupted temporarily.

Since there was no known study on dentistry students concerning COVID-19, studies related to infectious diseases were used to create the survey (*Alharbi et al., 2019*; *Lorosa et al., 2019*; *Myers et al., 2012*). We prepared the questions in three parts. In the first part, the participants were asked to supply demographic data (Course period, gender); in the second part, questions were asked about attitudes and knowledge (e.g.,the fear of infecting themselves or their environment while treating someone with COVID-19, on which internship worried them more, and on whether antibiotics are beneficial in COVID-19 treatment); and in the third part, questions were asked about biosafety procedures applied for COVID-19 (e.g., individual and infection control precation).

**Table 1** Distribution of the students according to gender and course period.

| Students | | N | (%) | N |
|---|---|---|---|---|
| Gender | Male | 165 | (46.4%) | 355 (100%) |
| | Female | 190 | (53.6%) | |
| Course period | Clinical | 113 | (31.9%) | 355 (100%) |
| | Preclinical | 242 | (68.1%) | |

## Data analysis

SPSS 21.0 for Windows was used to make a statistical analysis of the data. Descriptive statistical methods and a chi-square test were employed. The significance level was set at $p < 0.05$.

## RESULTS

Of the 355 students, 190 (53.6%) were female and 165 (46.4%) were male; 242 (68.1%) were first, second, and third grade students doing their preclinical education, and 113 (31.9%) were fourth and fifth year students doing their clinical rotations (Table 1).

Table 2 shows the knowledge and attitudes towards COVID-19 as shown by their questionnaire responses. In respect of COVID-19 treatment, 80% of participants responded "no" to the question of whether antibiotics are beneficial, because COVID-19 is a viral infections and antibiotics may only benifical serious co-infections status. There were no significant differences in terms of gender or clinical-versus-preclinical students for these questions.

A quarter (25.1%) responded yes to the question of whether a lecture or seminar-like information had been given in their school about COVID-19. The difference between preclinical students (19.8%) and clinical students (36.3%) was statistically significant ($p = 0.01$) (Table 2). Figure 1 shows the sources of information about the disease and virus. To the question of where they gained information about COVID-19, 75.8% indicated the websites or social media accounts of professional organizations, such as the Ministry of Health, Dental Association, and WHO, 21.9% gave information meetings held in institutions, 29.2% gave published scientific articles, 41.4% physicians' individual websites or social media accounts, 60.1% social media accounts, like Instagram, Facebook, and Twitter, 64.8%, television and radio programs, and 65.3% communication groups, such as Whatsapp or Line.

The question about whether their experiences related to COVID-19 affected them psychologically received "yes" response from the majority (74.9%) of participants. The difference between females (80.5%) and males (68.5%) was statistically significant ($p = 0.02$); the difference between preclinical students (70.2%) and clinical students (85%) was also statistically significant ($p = 0.01$) (Table 2).

To the question about whether they were afraid of being infected by the COVID-19 virus since they were entering a profession that works very closely with other healthcare workers and patients: total of 82.3% of the participants responded "yes". For males, this figure was 74.5% and for females 88.9%, a statistically significant difference ($p = 0.001$); for

Peer*J*

**Table 2  Knowledge and attitudes of students about COVID-19.**

| Questions | | Male (n = 165) % | Female (n = 190) % | P value | Preclinical (n = 242) % | Clinical (n = 113) % | P value |
|---|---|---|---|---|---|---|---|
| Have you been informed about COVID-19 in your faculty like lectures or seminars? | Yes | 25.5 | 24.7% | .486 | 19.8% | 36.3% | .001* |
| | No | 74.5% | 75.3% | | 80.2% | 63.7% | |
| Have you asked questions such as high fever, dry cough or travel abroad while taking an anamnesis from your patients in the last 3 months? | Yes | 13.9% | 16.8% | .273 | 17.4% | 11.5% | .102 |
| | No | 86.1% | 83.2% | | 82.6% | 88.5% | |
| Did COVID-19 negatively affect your psychology? | Yes | 68.5% | 80.5% | | 70.2% | 85% | |
| | No | 19.4% | 6.8% | .002* | 13.2% | 11.5% | .001* |
| | Undecided | 12.1% | 12.6% | | 16.6% | 3.5% | |
| Are you afraid of becoming infected with COVID-19 as a healthcare professional working at close range with the patient? | Yes | 74.5% | 88.9% | | 77.6% | 92% | |
| | No | 15.2% | 5.3% | .001* | 12% | 5.3% | .002* |
| | Undecided | 10.3% | 5.8% | | 10.4% | 2.7% | |
| Are you afraid to infect any relatives or people around you in terms of COVID-19 because you are a healthcare worker working very close to the patient? | Yes | 91.6% | 94.2% | | 90.5% | 98.2% | |
| | No | 4.2% | 3.7% | .485 | 5.4% | 0.9% | .012* |
| | Undecided | 4.2% | 2.1% | | 4.1% | 0.9% | |
| Would you hesitate to treat a patient who came to dental treatment after getting over and recovering from COVID-19 infection? | Yes | 37.6% | 50% | | 40% | 53% | |
| | No | 47.2% | 27.4% | .001* | 38.9% | 31.9% | .066 |
| | Undecided | 15.2% | 22.6% | | 21.1% | 15.1% | |
| Do you think that after your COVID-19 pandemic, you will be more careful in your standard measures regarding contamination in your patients? | Yes | 77.6% | 88.9% | | 80.2% | 91.2% | |
| | No | 4.8% | 1.6% | .012* | 3.7% | 1.7% | .033 |
| | Undecided | 17.6% | 9.5% | | 16.1% | 7.1% | |
| With the COVID-19 outbreak, did you regret that you chose the dentistry profession? | Yes | 9.7% | 8.9% | | 7.9% | 12.4% | |
| | No | 78.8% | 74.7% | .426 | 78.1% | 73.5% | .382 |
| | Undecided | 11.5% | 16.4% | | 14% | 14.2% | |
| Do antibiotics benefit in the treatment of COVID-19? | Yes | 3.6% | 3.7% | | 2.2% | 5.1% | |
| | No | 83% | 77.4% | .357 | 76.4% | 83.5% | .133 |
| | I don't know | 13.4% | 18.9% | | 21.4% | 11.4% | |
| Can a mother diagnosed with COVID-19 breastfeed her child? | Yes | 15.2% | 12.1% | | 11.6% | 14.7% | |
| | No | 49.0% | 54.2% | .559 | 53.3% | 49.% | .285 |
| | I don't know | 35.8% | 33.7% | | 35.1% | 35.6% | |

**Notes.**
*Chi-square test *p* < 0.05.

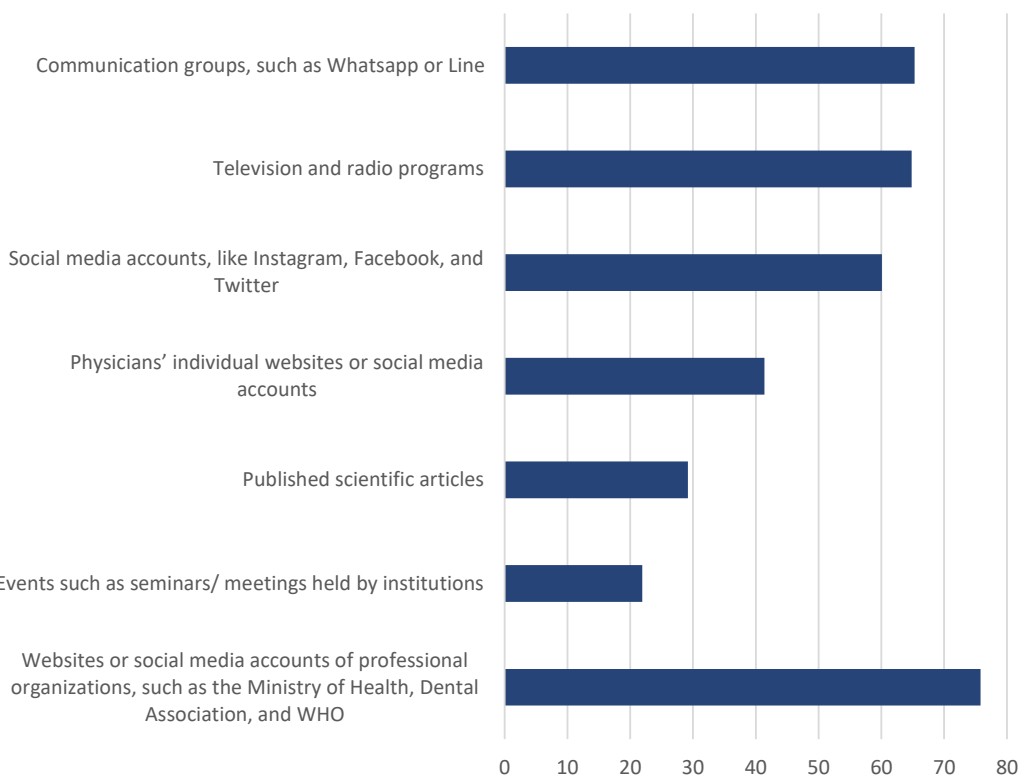

**Figure 1 Sources about information about COVID-19.**

preclinical students, it was 77.6% and clinical students 92%, again, a statistically significant difference ($p = 0.002$) (Table 2).

To the question of whether they were afraid of infecting relatives or people around them with the COVID-19 virus because of their profession: 93.0% of the participants responded "yes". There was no significant difference by gender. Hence the difference between the preclinical (90.5%) and clinical (98.2%) students was statistically significant ($p = 0.012$). A total of 44.2% of the participants replied "yes" to the question of whether they would hesitate to treat a patient who came to dental treatment after recovery from COVID-19 infection, comprising 50% of the females and 37.6% of the males, which was statistically significant ($p = 0.001$). The difference between preclinical (40%) and clinical (53%) students for this measure was not significant ($p = 0.066$) (Table 2).

Regarding whether they thought that after the COVID-19 pandemic, they would be more careful in their standard measures related to contamination of their patients. 77.6% of females responded "yes" and 88.9% of males responded "yes", which was statistically significant ($p = 0.012$). The proportion for preclinical students was 80.2% and for clinical students 91.2%, which also constituted a significant difference ($p = 0.033$) (Table 2).

In the wake of the onset of the COVID-19 pandemic, 74.7% of the participants responded "no" to the question if they regretted having chosen the dentistry profession. There was

**Table 3** Demonstrates in which clinical rotation students are more concerned about COVID-19.

| Clinical rotation | Preclinical $(n = 242)\%$ | Clinical $(n = 113)\%$ | P value |
|---|---|---|---|
| Oral and Maxillofacial Surgery | 35.1% | 4.5% | |
| Endodontics | 24.8% | 40.7% | |
| Periodontology | 14.5% | 16.8% | |
| Prosthodontic | 13.2% | 23% | .001[*] |
| Oral Diagnosis and Radiology | 2.1% | 7.9% | |
| Restorative dentistry | 6.6% | 7.1% | |
| Pediatric Dentistry | 2.1% | 0.8% | |
| Orthodontics | 1.7% | 0% | |

Notes.
[*]Chi-square test $p < 0.05$.

no significant gender or preclinical and clinical student difference ($p = 0.426$, $p = 0.382$) (Table 2).

Table 3 demonstrate that the students' concern about which clinical rotation are more dangerous for COVID-19 contamintaion. Regarding the clinical rotation that worried participants most, 29.9% gave endodontics as their answer, 25.1% oral and maxillofacial surgery, 16.3% prosthesis, 15.2% periodontology, 6.8% restorative dentistry, 3.9% oral diagnosis and radiology, 1.7% pedodontics, and 1.1% orthodontics. There was a significant difference between the preclinical and clinical students ($p = 0.001$).

Figure 2 shows the individual measures taken by students against COVID-19. To the question of which individual measures they were taking against COVID-19, the wearing of gloves and of a mask received responses of 33.8% and 44.4%, respectively; frequent hand-washing was 93% and use of cologne, wet wipes, and hand disinfectant 84.9%; not entering public areas was 93.3%, not having physical contact (handshaking, kissing, etc.) 88.8%, and frequent ventilation of the environment 78.2%, while changing clothes and taking a shower upon arrival home were 65.3% and 33.2%, respectively; 2.7% of the respondents indicated that they did not do anything extra.

Figure 3 shows the precautions taken by clinical students for themselves in clinical rotation. These participants responded to the above question on which measures they were taking in the following proportions—use of gloves: 100%, mask: 100%, ffp3/n95 mask: 11.5%, face protective shield: 73.6%, safety glasses: 37.1%, bone: 49%, disposable box: 16.8%, frequent hand-washing: 90.2%, and frequent hand antiseptic: 86.7%.

Figure 4 shows the precautions taken by clinical students regarding COVID-19 with the patient during dental treatment. To the question of which measures for COVID-19 they were taking in this situation, the responses were as follows. Before the procedure, 73.4% asked whether the patient had symptoms, such as fever, cough, or shortness of breath, 14.2% measured the patient's fever, 15.9% applied a rubber dam, 17.7% rinsed the mouth with a mouthwash containing chlorhexidin, 1.77% rinsed the mouth with a mouthwash containing 1% hydrogen peroxide content, 42.4% used a strong saliva absorber during the procedure, 12.3% avoided aerosols and processes that would create droplets, preferring to use hand tools instead of an aerator, cavitron, or micromotor, 24.7% postponed

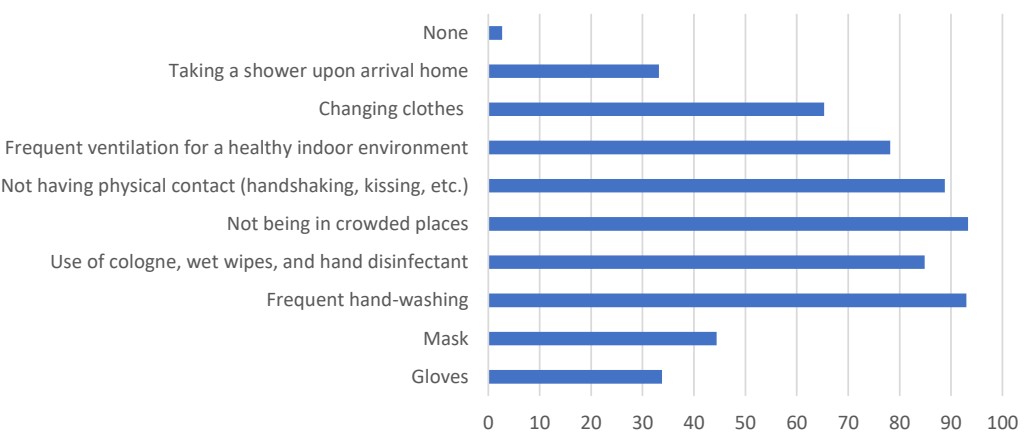

**Figure 2** Individual measures taken by dental students against COVID-19 in their daily lives.

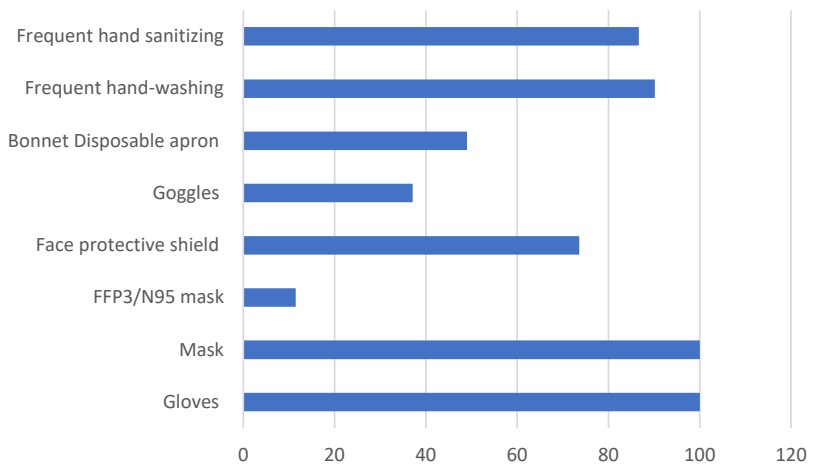

**Figure 3** Precautions taken by clinical students for themselves in clinical rotation related to COVID-19.

appointments of potentially infected patients for at least 14 days, and 8.8% stated that they did nothing.

# DISCUSSION

Dentists, dental students, and auxillary staff are at more risk of encountering pathogens transmitted through blood or other body fluids than the normal population (*Al-Maweri et al., 2015*). The key to reducing and preventing contamination of various microorganisms is strict adherence to infection control procedures. Thus, the knowledge about and attitudes towards infectious diseases of students who have started patient treatment procedures in the clinic are very important. Less experienced students are likely to be more susceptible to the risk of infection diseases (*Singh & Purohit, 2011*).
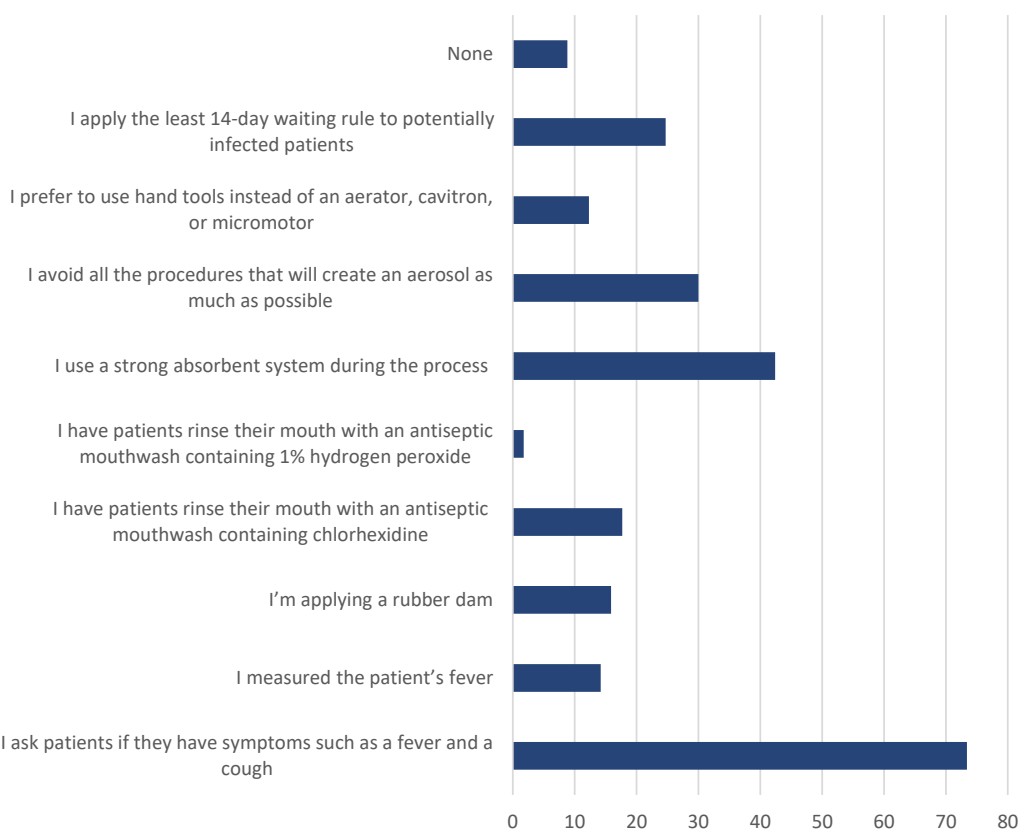

**Figure 4** Precautions taken with the patient regarding COVID-19 while dental procedures.

There are many studies investigating the knowledge levels and attitudes of dental students about infectious diseases (*Al-Maweri et al., 2015*; *Al-Shamiri et al., 2018*; *Alharbi et al., 2019*; *Karcilioglu, 2020*; *Lorosa et al., 2019*; *Myers et al., 2012*). COVID-19 is a very new disease that has spread rapidly and about which information is limited. To our knowledge, no study has yet been made related to COVID-19 and dental students. This study investigated the knowledge, attitudes and practices regarding COVID-19 of preclinical and clinical dental students at Firat University, Turkey.

COVID-19 transmission routes are through direct contact and airborne droplets, including aerosol delivery (*Ge et al., 2020*). Most of the treatments in dentistry produces droplets and/or aerosols that can cause infection. Students, especially those with limited clinical experience, should be very careful about infectious diseases, to protect themselves and for their patients and the employees. To prevent cross-infection in dentistry, standard measures should be taken such as disposable surgical cap, disposable surgical mask, white coat, safety glasses or face protection, use of disposable latex or nitrile gloves (*Peng et al., 2020b*). In the treatment procedures of patients, the use of mouthwash (1% hydrogen peroxide or 0.2% povidone iodine), use of "rubber-dam" (performing dental practice with hand tools in a way that does not create an aerosol in conditions where it is not possible to use "rubber-dam") is recommended (*Samaranayake & Peiris, 2004*). The use of rubber
dam results in a significant reduction in the microbial content of air turbine aerosols produced during operative procedures, thereby reducing the risk of cross-infection in the dental practice (*Ahmad, 2009*). It is also recommended to avoid splashing or aerosole processes using extra traction measures such as the use of a high-traction saliva absorber and, if possible, the use of an aerator with an anti-retraction valve (*Samaranayake & Peiris, 2004*). Strict disinfection measures should be taken in clinics, the environment should be ventilated after treatment, and the areas of contact (unit, reflector..) should be disinfected (*Peng et al., 2020b*).

In this study, 74.9% of the students reported that there was no course approved or seminar-like information about COVID-19 provided in their school. As COVID-19 very quickly became a pandemic, one may suggest that dental schools may provide extra informative about this disease in their approved curriculum. Some three quarters (75.8%) of the participants had received information about COVID-19 from the websites or social media accounts of professional organizations, such as the Ministry of Health, the Dental Association, and the WHO, with a fifth (21.8%) gaining information from meetings in institutions. In previous studies, the most important information source for students was those of mass communication (television, newspapers, and magazines) (*Gökengin et al., 2003*; *Opt & Loffredo, 2004*; *Ungan & Yaman, 2003*). Nowadays, we can easily argue that social media accounts have replaced the mass media. The experience we have acquired with COVID-19 suggests that after outbreaks and with the emergence of an epidemic, school administrations should definitely hold lectures and/or informative meetings for their students. The awareness of students regarding scientific articles should be increased.

In this study, 80% of students stated that antibiotics would not be useful in COVID-19 treatment, presumably because they knew that it is a viral disorder. While this 80% rate was satisfactory, we think that it should have been higher.

Only 11.5% of the clinical students asked their patients if they were treated in the last three months and if they had a high fever or dry cough or traveled abroad. This indicates a need to inform students as soon as possible about diseases following outbreaks and for them to provide the necessary information in their history.

To the question on whether COVID-19 affected them in a negative way psychologically, according to the participants responses a difference between females (80.5%) and males (68.5%) that was statistically significant ($p = 0.02$). In a study conducted at a medical school in China, it was shown that the psychological status of female and male students there was similarly affected by the COVID-19 outbreak (*Cao et al., 2020*). Physiological sensations related to stress, social phobia, depression, panic and fear are widespread in women, and these may be related to anxiety. It may be said that women are more negatively affected by the stressful periods than men and that anxiety is positively related to this psychological condition (*Yildirim et al., 2017*). Already, under normal conditions, it has been shown in many studies that female dental students are more stressed than their male counterparts. The reason for this situation is thought to be related to the fact that women feel stress more intensely in general while men hide their anxiety (*Divaris et al., 2013*; *Jowkar, Masoumi & Mahmoodian, 2020*). Further, there were differences between the proportions of preclinical and clinical students who thought that the disease had a

negative psychological impact (70.2% and 85%, respectively). They were afraid of being infected with COVID-19 (77.6% and 92%, respectively), which were statistically significant ($p = 0.02$ and $p = 0.04$, respectively). The majority of the participants (93.0%) responded " yes" to the question of whether they were afraid of infecting someone around them with COVID-19. The difference between the preclinical (90.5%) and clinical (98.2%) students was also statistically significant ($p = 0.012$). These results may be explained by the fact that clinical students are in contact with patients during dental treatment and so the risk of infection transmission is higher for clinical than for preclinical students. Dental clinical students have increasing patient contact during their education and clinical years which may put them in an increased risk of cross infections (*Milward & Cooper, 2007*).

To the question of whether they would hesitate to give dental treatment to a patient who had COVID-19: 44.2% of the participants responded "yes", 36.6% responded "no" and 19.2% responded undecided. Studies have shown that as the knowledge level of dentistry faculty students increases, so does their willingness to treat patients with infectious diseases (*Aggarwal & Panat, 2013*; *Sadeghi & Hakimi, 2009*). Prejudice against such patients in respect of COVID-19 may be prevented by providing students with appropriate clinical rotation. N95 mask should be used in addition to standard measures during emergency dental procedures for the person who has infection suppression or infected. Surgical masks are primarily used to stop the wearer from spreading microbes while coughing, sneezing or talking. The most important difference between respirators and surgical masks that they do not have a tight seal due to the gaps in the surgical masks. When it comes to infection control in the field of health, N95 masks with particle filters are preferred.These include three layers, being hypoallergenic, forming a liquid barrier, being tear resistant, and providing 99%bacteria and 95% particle filtration (*Ahmad, 2009*).

Regarding the question of which clinical rotation worried them more, the clinical students were most concerned about endodontics (40.7%), prosthetics (23%), and periodontology (16.8%) while pre-clinical students were most concerned about oral and maxillofacial surgery (35%), endodontics (24.7%), and periodontology (14.5%). We think that this is due to the better awareness of clinical students about aerosols and droplets. Infected children with COVID-19 have relatively mild clinical symptoms compared to infected adults. No deaths in the pediatric population was reported at the time of the study (*Sun et al., 2020*). We think that the very low rate of pedodontics (1.7%) was related to this situation.

For the question about, after the COVID-19 pandemic, students were more careful in their standard measures regarding patient contamination, the majority (83.7%) of participants responded "yes". COVID-19 appears to have increased the awareness of students of the risks of infectious diseases.

Three quarters (74.7%) of participants in the study responded "no" to the question of whether they regretted having chosen dentistry. This result may be explained by the assessment that as student knowledge about infectious diseases increases, such regrets become meaningless.

Many individual measures were taken by students in daily life related to COVID-19, the foremost among which were not entering public areas (93.3%), frequent hand-washing

(93%), avoiding physical contact (88.8%), and antiseptic use (84.9%). It is understood that these students were conscious of the individual measures they should take.

In respect of the results gained from questions on the precautions taken in clinical rotation and measures taken by students for their patients regarding COVID-19, the clinical students were found to be very careful about standard measures but less concerned about extra measures to be taken. This was probably due to their rotations in the week after the first cases in Turkey was reported.

## CONCLUSION

Dental students are inherently at high risk of exposure to infectious diseases. The emergence of COVID-19 brought new challenges and responsibilities to institutions providing dental education. In particular, students should be informed that special measures should be taken for asymptomatic carrier patients in addition to the standard measures. During the preclinical years, the knowledge of the students and appropriate attitudes to infectious diseases, especially pandemics, should be developed. This is extremely important in the fight against infectious diseases.

### Funding
The authors received no funding for this work.

### Competing Interests
The authors declare there are no competing interests.

### Author Contributions
- Osman Ataş conceived and designed the experiments, performed the experiments, prepared figures and/or tables, and approved the final draft.
- Tuba Talo Yildirim conceived and designed the experiments, performed the experiments, analyzed the data, prepared figures and/or tables, authored or reviewed drafts of the paper, and approved the final draft.

### Human Ethics
The following information was supplied relating to ethical approvals (i.e., approving body and any reference numbers):

The study was approved by the Fırat Üniversity Research Ethics Committee (2020/30-06).

### Data Availability
The raw data are available in Supplemental File.

### Supplemental Information
Supplemental information for this article can be found online at http://dx.doi.org/10.7717/peerj.9575#supplemental-information.

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
