# Peer review of "Evaluation of knowledge, attitudes, and clinical education of dental students about COVID-19 pandemic"

_PeerJ, doi:10.7717/peerj.9575_

## Round 0.1 · original submission · Major Revisions

Please respond to the reviewers' comments on a point by point basis in your rebuttal.

·

Basic reporting

- The introduction needs more details about COVID-19. There are reports about four different types of coronavírus, some infect humans and mammals and others, only birds. SARS and MERS viruses have caused serious respiratory diseases. SARS-CoV2 has a membranous structure of protein spines and penetrates cells through ACE2 cell receptors. Bats and humans are believed to be hosts of the virus and is speculated the presence of an intermediate host called pangolim(squamous anteater) mammal that inhabits tropical areas of Asia and Africa.

- The English language should be improved in all document.

Experimental design

- Methods are simple but well described

Validity of the findings

- Tables and graphics can be improved, considering aesthetic aspects

Additional comments

- I suggest that on the topic “discussion” the authors:
* explain the difference between surgical masks and N95 respirators. Also on this topic, they should enfasize the use of rubber dam in endodontics, which can minimize the spraid of virus.
*Talk about preventive measures to control and minimize infection in dental pratice

- The bibliografic references should be reviewed and padronized

·

Basic reporting

•Basic reporting needs to improve as there are grammatical errors throughout the manuscript. It is advised to do a grammar check before submitting revision.

•In the structured abstract background could be improved as to why your team is doing this study? Background should be focused on the importance of dental students equipping themselves with COVID-19 practice guidelines and precautions. Objective can be added as heading or modify the objective to be a continuation of the background. Results section in abstract can be concised, avoid percentages and statistically significant reporting with p value.

•Structuring of results could be improved by creating sections. In the questionnaire report knowledge questions first, then attitude and then practice. Follow the same in the results. Discussion could also be structured better a paragraph should address one issue and everything related can be clubbed in that.

•For the figures please add horizontal and vertical axis titles. Refrain from using words like our students in the figure titles.

Experimental design

•The article is a cross-sectional study within the scope of journal. The gap in knowledge is addressed, but the rationale for gender comparison is unclear.

•The methods are described clearly with appropriate statistical tests application.

•Replace participation rate with response rate as it is more ideal for online surveys. Avoid terms like negative response as this can confuse readers.

•The question of antibiotics usage is not appropriate as the disease in question is a virus and antibiotics are only advised for co-infections. Justify why there is a need to include benefits of antibiotics in the questionnaire.

Validity of the findings

• The study validity can be improved by providing the current status of dental students, whether they are suspended temporarily during the outbreak or are currently working as per the government regulations in Turkey. The study results can be useful to similar dental institutions.

Additional comments

•The study could be added to the evidence base and is likely to get citations. But grammatical errors and structuring of manuscript needs to be improved

---

## Round 0.2 · accepted · Accept

The revisions made in your manuscript are acceptable to the reviewers. Your manuscript is now suitable for publication.

·

Basic reporting

The authors did the review properly and the paper can be published.

Experimental design

The authors did the review properly and the paper can be published.

Validity of the findings

The authors did the review properly and the paper can be published.

Additional comments

The authors did the review properly and the paper can be published.

·

Basic reporting

no comment

Experimental design

no comment

Validity of the findings

no comment